# New Photochromic α-Methylchalcones Are Highly Photostable, Even under Singlet Oxygen Conditions: Breaking the α-Methyl Michael-System Reactivity by Reversible Peroxybiradical Formation

**DOI:** 10.3390/molecules26030642

**Published:** 2021-01-26

**Authors:** Axel G. Griesbeck, Banu Öngel, Eric Brüllingen, Melissa Renner

**Affiliations:** Department of Chemistry, University of Cologne, Greinstr. 4, 50939 Köln, Germany; banu.oengel@gmail.com (B.Ö.); eric.bruellingen@uni-koeln.de (E.B.); melissa.renner@gmx.de (M.R.)

**Keywords:** chalcones, photochromic compounds, singlet oxygen, reactivity, E/Z-isomerization

## Abstract

The α-methylated chalcones **7a**–**7e** behave as P-type photochromic substances with photo-stationary states (PSS) as high as 15:85 when irradiated at 350 nm. These compounds are easily accessible in pure E-configuration by aldol condensation or by oxidative coupling/elimination. The α-methyl groups make these compounds potentially reactive with singlet oxygen following the gem-rule that predicts ^1^O_2_ regioselectivity. Even after long irradiations times in the presence of the singlet oxygen sensitizer tetraphenylporphyrin (TPP) and oxygen, however, no oxygenation products were detected. Under these conditions, all substrates were converted into 9:1 E/Z-mixtures despite the use of low-energy light that does not allow direct or sensitized excitation of the substrates **7**. Additionally, chalcone **7a** reduced the singlet oxygen reactivity of the tiglic ketone **3a** by about a factor of two, indicating substantial physical quenching of singlet oxygen by the α-methylated chalcones **7a**–**7e**. Thus, a singlet oxygen-induced E/Z-isomerization involving 1,2-dioxatetra-methylene biradicals that leads to triplet oxygen and thermodynamic E/Z-mixtures is postulated and supported by quantum chemical (DFT)-calculations.

## 1. Introduction

Singlet oxygen (^1^∆_g_ − ^1^O_2_) is a highly reactive and structurally simple oxygenation reagent. It is easily generated from air oxygen (ground-state electronic triplet state) by energy transfer from a vast number of electronically excited state dye molecules [1,2,3,4]. Because it is such an easily available oxidation and oxygenation (O versus O_2_-transfer) reagent, ^1^O_2_ has found numerous applications in organic synthesis [5,6,7,8,9]. Besides heteroatom oxidation, the following pericyclic reactions with unsaturated organic substrates are especially relevant: 2 + 2-cycloaddition with electron-rich monoalkenes, 4 + 2-cycloaddition with 1,3-dienes, and ene-reaction with monoalkenes or polyenes carrying allylic hydrogens [10,11,12]. Especially, the latter process has huge synthetic potential because allylic oxidation is one of the synthetically most relevant steps that allows the generation of allylic hydroperoxides, alcohols, epoxides, and many more [13]. In contrast to other oxidation reagents, singlet oxygen has two severe limitations that make applications cumbersome. Firstly, as an excited state, ^1^O_2_ has an intrinsic lifetime that is solvent-dependent and ranges from some nanoseconds to hundreds of microseconds in solution and minutes/hours in the gas phase [14,15,16]. Thus, even very fast chemical reactions can proceed quite slowly if solvents are used with short ^1^O_2_ lifetimes [17,18]. Secondly, many substrates physically quench ^1^O_2_, i.e., deactivate the excited state without being chemically changed [4]. The interplay between chemical and physical quenching and intrinsic unimolecular decay, characterized by the rate constants k_r_, k_q_, and k_d_, is thus crucial for the prediction of reactivity (chemical vs. physical quenching) and selectivity (chemo-, regio-, and stereoselectivity). Alkenes with allylic hydrogen atoms are robust substrates for ene-chemoselectivity with several guidelines for the prediction of preferred hydrogen transfer: *cis*-effect [19], *gem*-effect [20,21], or *large-group* effect [21,22]. In this context, many natural products and likewise model compounds for natural products are described as “antioxidants”. Especially, electron-rich unsaturated molecules such as polyenes and (poly)aromatics are highly reactive with singlet oxygen, either in a physical quenching way or by chemical reaction [23,24]. Thus, the predictability of chemical reactivity for any potential singlet oxygen substrate is impossible without knowledge of physical quenching pathways such as spin catalysis or electron or energy transfer. Every new class of organic substrates therefore constitutes an unclear starting situation with respect to the chemical behavior under singlet oxygen conditions. We were especially interested in chalcones in the course of our investigations of naturally occurring antioxidants [6,25,26] and designed new potentially reactive and chemically modified chalcones.

## 2. Results and Discussion

Chalcones are aromatic Michael ketones, which are accessible in a wide range of colours due to their extended conjugated π systems [27,28]. These compounds are known as biologically and pharmacologically active molecules with diverse properties [29,30]. In order to become applicable for singlet oxygen Schenck-type ene reaction, α-methylated chalcones were chosen for the investigation. A large family of corresponding α-methylated α,*β*-unsaturated ketones, aldehydes, esters, amides, and carboxylic acids have already been described in the literature as reactive in ^1^O_2_ ene reactions with high hydrogen transfer selectivity from these α-methyl groups (Scheme 1, termed the ^1^O_2_ “gem effect” in the literature). Thus, chalcones that are α-unsubstituted cannot undergo ene reactions because the lack of allylic hydrogens and α-methyl groups introduced enables hydrogen abstraction from this position that corresponds to the regiochemically preferred position in the “gem effect” [20,21].

The basic background reaction concept is depicted in Scheme 2: the tiglic ketone **3a** reacts rapidly with singlet oxygen to give the allylic hydroperoxide **4a** with >98% regioselectivity. “Rapidly” is of course a relative term, the reactivity is about 1/20 of the non-Michael substrate cis-2-butene and in the magnitude of tiglic acid reactivity (k_r_(C_6_D_6_) = 7 × 10^3^ M^−1^s^−1^) [31]. The α-methyl group is thus essential for the overall reactivity, i.e., the crotonic ketone **3b** is unreactive and also the parent chalcone **4** likewise does not interact with singlet oxygen under the investigated reaction conditions [32].

In order to make chalcone derivatives more reactive with singlet oxygen, we decided to combine the structural features of chalcones with the α-methyl motif. Insertion of a α-methyl group into the chalcone structure **4** is possible by carbonyl coupling chemistry (Scheme 3). The derivatives **7b**,**d**,**e** were prepared by classical aldol reactions using 40% aqueous NaOH solution as base [33]. The yields (8–18%) for traditional aldol condensations were relatively poor for all substrates, obviously caused by the α-substitution. Compounds **7a** and **7c** were accessible by a tandem cross-dehydrogenative-coupling/elimination reaction in good yields [34].

Originally, the substituent pattern on the A- and B-rings of the chalcone skeleton were designed to shift the absorption bands into the visible range and to modify the reactivity of the alkenes. All compounds **7a**–**7e** that are synthetically produced as E-isomers in pure form from the above reactions are photochromic, i.e., are switchable into the Z-isomers by direct light absorption [35]. This photochromic behavior leads to stable E/Z-mixture compositions that are termed photostationary states (PSS), which are dependent on the wavelength applied for irradiation. For all compounds reported herein, the PSS are around 30/70 (E/Z) ± 10% over a wide excitation area (Scheme 4). The spectral changes accompanied with constant irradiation at 350 nm of the parent α-methyl chalcone **7a** are shown in Figure 1: the maximum absorption of the E-isomer E-**7a** (λ_max_ = 286 nm, ε^286^ = 18.977 L·mol^−1^·cm^−1^) is shifted to 246 nm with concomitant formation of the Z-isomer Z-**7a**, indicating a decrease in π conjugation (vide infra). This photochromic behavior involves electronic excitation of the photochromic compounds, which can be avoided by shifting the wavelength into the red part of the UV-vis spectrum or by adding a much stronger absorber to the solution. In case of the singlet oxygen experiments described in this paper, the red sensitizer meso-tetraphenylporphyrin (TPP) was used as sensitizer which, as a very strong absorber in the wavelength region between 350–450 nm, completely blocks the UV-absorption of the chalcones **7a**–**e**.

The synthetically aimed photooxygenation of the chalcones **7a**–**e** was performed always under conditions that exclude direct substrate excitation. As a standard protocol, stock solutions of CDCl_3_ with chalcone concentrations of 200 mM and sensitizer (TPP) concentration of 5 × 10^−4^ mol/l in CDCl_3_ were applied. The reaction vessel (5 mm quartz NMR tube) was irradiated with three white LEDs (30 W each) under constant oxygen exposure. The reaction control was performed directly with this reaction solution by NMR spectroscopy. The results for the chalcones **7a**–**e** for different reaction times (until no further changes were detected in the NMR spectra) are shown in Figure 2. In all cases, solely an E/Z photoisomerization of the double bond could be observed (E/Z ratio approx. 85:15, blue boxes) and there were no signs of photoxygenation products (allylic hydroperoxides or other peroxidic products). The E/Z ratio of the chalcones **7a**–**e** that was obtained after prolonged irradiation time (under singlet oxygen conditions) was nearly identical for all substrates in the error limit—in contrast to the photostationary equilibria that are obtained by direct excitation of these compounds. The presence of oxygen was crucial: compared to the isomerization of **7a** under oxygen in the presence of TPP (E/Z = 89:11), the ratio under argon/TPP resulted in a E/Z ratio of 97:3. This accounts for the fact that the E/Z-ratios originating from direct substrate excitation (leading to PSS) and the E/Z-ratios originating from singlet oxygen experiments originated from different pathways.

These results were striking because they indicate that there is indeed an interaction between singlet oxygen and the substrate that is not accompanied by the formation of oxygenation products. Accordingly,, singlet oxygen was expected to be quenched by the α-methyl chalcones. Both energy and electron transfer quenching appear to be unlikely: the triplet energies of chalcones (~2.2 eV for the unsubstituted compound) [35] are more than twice that of singlet oxygen and one-electron oxidation of Michael ketones appears also to be energetically unfeasible. Likewise, energy transfer from the electronically excited sensitizer TPP is unfeasible; the triplet energy of TPP is lower than 1.5 eV [36] and is thus not be able to excite the chalcone derivatives by energy transfer.

In order to display the quenching properties of **7a**, the photooxygenation of (E)-3-methylpent-3-en-2-one (**3a**) [37] was investigated in the absence and presence of **7a** (Scheme 5). For this purpose, two experiments were performed in parallel: **3a** was irradiated with white LEDs (30 W) in a CDCl_3_ solution (5 × 10^−4^ M in TPP) in an NMR tube and in parallel, and **3a** was exposed together with chalcone **7a** (identical concentrations of 0.2 M) for 20 h with 1 h/5 h steps and NMR analyses. From the linear parts of the comparison of the two conversion, it is apparent that **3a** in the absence of **7a** reacts about 1.4 times faster with ^1^O_2_.

Thus, the presence of chalcone **7a** has a pronounced retarding effect on the chemical reactivity of a singlet oxygen acceptor without being consumed, i.e., physical quenching of singlet oxygen. The latter process is accompanied by double bond isomerization of the central CC chalcone bond. This behavior was also reported for the acyclic diene **8**, which reacts with singlet oxygen but is also Z/E-isomerized in a competing addition/elimination process converting (= quenching) singlet oxygen to triplet oxygen [38]. Likewise, singlet oxygen reactions with electron-rich α-methylated 1,3-diene carboxylic esters such as **10**, which proceed via a vinylogous ene reaction, were computationally analyzed as proceeding via (peroxy)biradical intermediates [39]. Both intermediate-induced physical quenching is, however, accompanied by chemical reactions, i.e., product formation. Seemingly, this is not the case for the chalcones described herein (Scheme 6) that find no further product pathway.

The assumption of intermediary singlet 1,4-biradicals is supported on one hand by the photooxygenation experiments that show ^1^O_2_-induced E/Z-isomerization. Energy or electron transfer quenching can be excluded due to energetic considerations. Quantum chemical (DFT) computations (TPSS-D3(BJ)/def2-TZVP) of the proposed reaction profile (Figure 3) reveal that the E-isomer of **7a** is preferentially in its *s-trans* configuration avoiding additional Ph/Me steric strain. This is in accordance with solid-state measurements for donor-substituted α-methyl chalcones [40]. The Z-isomer of **7a** is distorted, resulting into two atropisomeric structures, which are identical in energy. This is a remarkable spin-off product of the DFT-calculations that showed only two degenerated minima for the Z-**7a** structure revealing that s-trans/s-cis assignment is no longer possible for the Z-isomers. The strong Ph/Ph-π-distortion in Z-**7a** is in excellent agreement with the 50 nm blue-shift observed during E→Z-photoisomerization. The very low energy difference of 0.5 kcal/Mol between E-**7a** and Z-**7a** corresponds roughly to a 80:20 E/Z-equilibrium that is in accord with the measured values.

The perepoxide-like transition state **TS** [41] in the interaction with singlet oxygen is 17.6 kcal/Mol above the substrate energy of E-**7a** and the singlet biradical **BR^1^** follows only one kcal/Mol lower in energy. From this **BR^1^**, no product formation (beside the energetically unfavourable 1,2-dioxetane formation) is possible and re-formation of a thermodynamic equilibrium of E/Z-substrates with elimination of triplet oxygen results. A second singlet biradical **BR^2^** can be competitively formed with relatively similar energy but apparently does also not contribute to any product formation. The calculated energy differences between the singlet and triplet biradical pathways in Figure 3 are minimal (<0.5 kcal/Mol, not included in Figure 3), and thus intersystem crossing from the initial biradicals **BR^1^**/**BR^2^** into the triplet manifold is expected to occur rapidly. The computational analysis of these spin-forbidden processes can be performed by minimum energy crossing point (MECP) calculations [42,43].

Thus, this energy profile supports the assumption of a quenching mechanism that involves covalent bonding of the quencher with singlet oxygen and subsequent decomposition into triplet oxygen [44]. A similar process has already been described by Garavelli et al. for highly conjugated carotenoids as quenchers in competition with the well-known energy-transfer quenching [23].

## 3. Conclusions

The α-methyl chalcones **7a**–**7e** are potential singlet oxygen acceptors due to their electronic structure and the reactive α-methyl allylic hydrogens. Even after prolonged irradiations, however, these compounds show no trace of oxygenation products. When the chalcones are excited by UV-A-irradiation, E/Z-isomerization occurs and photostationary E/Z-mixtures ranging between 44:56 and 26:74 are formed. Under singlet oxygen conditions, the pure E-isomers **7a**–**7e** are also isomerized to nearly identical E/Z-mixtures of 90:10 (±1). Thus, direct electronic excitation or energy transfer cannot be involved in the singlet oxygen induced photoisomerization. That singlet oxygen is involved in this process was demonstrated by a competitive kinetic quenching experiment with the reactive tiglic ketones as ^1^O_2_ acceptor. DFT calculations suggest that singlet oxygen and the chalcones 7 react with formation of 1,2-dioxatetramethylene biradicals that decompose rapidly to triplet oxygen and thermodynamic E/Z-mixtures.

## 4. Experimental Section

Infrared spectra were obtained using a Perkin-Elmer 1600 series FTIR spectrometer (Perkin-Elmer, Walluf, Germany) and are given in cm^−1^ units. ^1^H-NMR spectra were recorded on a Bruker Avance 300 or on a Bruker Avance 500 spectrometer (Bruker, Ettlingen, Germany) instruments operating at 500 MHz. Chemical shifts are reported as *δ* in ppm and the coupling constants *J* in Hz units. In all spectra, the solvent peaks were used as the internal standard. Solvents used were CDCl_3_ (*δ* = 7.24 ppm) and MeOH-*d*_4_ (*δ* = 3.35, 4.78 ppm). Splitting patterns are designated as follows: s, singlet; d, doublet; t, triplet; q, quartet; m, multiplet; br, broad; the ^13^C-NMR spectra were recorded either on a Bruker Avance 300 spectrometer instrument operating at 75 MHz or on a Bruker Avance 500 spectrometer instrument operating at 125 MHz. High-resolution mass spectra (HR-MS) were recorded on a Finnigan MAT 900 spectrometer (Scientific Instrument Services, Ringoes, NJ, USA) and measured for the molecular ion peak (M^+^). IR spectra were obtained on a Si crystal Fourier-Transform spectrometer by Thermo Scientific (Nicolet 380 FT-IR). Absorption spectra were recorded on a Perkin-Elmer Lambda 35. The samples were placed into quartz cells of 1 cm path length. All samples were measured in a concentration of 10^−5^ M in acetonitrile.

General procedure (GP): 1.0 eq. propiophenone derivative and 1.0 eq. aldehyde were added in either methanol or ethanol, and approximately 3.0 eq. piperidine and acetic acid were added. The reaction solution was heated for reflux. After completion of the reaction, the solvent was removed under reduced pressure and the residue was dissolved in water and ethyl acetate. The solution was extracted three times with 3 × 20 mL ethyl acetate and washed with brine, and the combined organic phases were dried over MgSO_4_. After removing the solvent under reduced pressure, the crude product was purified by column chromatography [33]. Chalcones **7a–c** are literature-known compounds and were synthesized by condensations reactions with yields of 57%, 13%, and 77%; **7d** and **7e** (18%, 8%) are literature-unknown.

*(E)-2-methyl-1,3-diphenylprop-2-en-1-one* (**7a**): ^1^H-NMR (300 MHz, CDCl_3_): *δ* (ppm): 7.74 (d, *J* = 7.5 Hz, 2H, H-3, H-3′), 7.57–7.30 (m, 8H, H-1, H-2, H-2′, H-10, H-10′, H-11, H-11′, H-12), 7.18 (s, 1H, H-8), 2.27 (s, 3H, H-7), ^13^C-NMR (75 MHz, CDCl_3_): *δ* (ppm): 199.5 (s, C-5), 142.3 (d, C-8), 138.6 (s, C-4), 137.0 (s, C-6), 135.9 (s, C-9), 131.8 (d, C-1), 129.8 (d, C-3, C-3′), 129.6 (d, C-11, C-11′), 128.7 (d, C-2, C-2′), 128.6 (d, C-12), 128.3 (d, C-10, C-10′), 14.5 (q, C-7), FT-IR (ATR): *ṽ* (cm^−1^) = 1722 (m), 1596 (m), 1283 (m), 1252 (s), 1183 (w), 1169 (m), 1110 (m), 1015 (m), 771 (m), 611 (w), 602 (m); UV/vis (10^−5^ M in CH_3_CN): λ_abs_ (max) = 254 nm (ε = 13.152 M^−1^cm^−1^), 286 nm (ε = 18.977 M^−1^cm^−1^).

*(E)-3-(4-methoxyphenyl)-2-methyl-1-phenylprop-2-en-1-one* (**7b**): ^1^H-NMR (300 MHz, CDCl_3_): *δ* (ppm): 7.72–7.69 (m, 2H, H-3, H-3′), 7.52–7.49 (m, 1H, H_ar_), 7.46–7.38 (m, 4H, H_ar_), 7.15 (s, 1H, H-8), 6.93 (d, *J* = 8.78 Hz, 2H, H-11, H-11′), 3.83 (s, 3H, H-13), 2.28 (d, _J_ = 1.41 Hz, 3H, H-7), ^13^C-NMR (75 MHz, CDCl_3_): *δ* (ppm): 199.7 (s, C-5), 160.1 (s, C-12), 142.8 (d, C-8), 139.0 (s, C-4), 135.0 (s, C-6), 131.5 (d, C_ar_), 131.5 (d, C_ar_), 129.5 (d, C_ar_), 128.5 (s, C-9), 128.2 (d, C_ar_), 114.1 (d, C-11, C-11′), 55.4 (q, C-13), 14.5 (q, C-7), FT-IR (ATR): ṽ (cm^−1^) = 1722 (m), 1596 (m), 1283 (m), 1252 (s),1183 (w), 1169 (m), 1110 (m), 1015 (m), 771 (m), 693 (w), 602 (m); UV/vis (10^−5^ M in CH_3_CN): λ_abs_ (max) =231 nm (ε = 12.688 M^−1^cm^−1^), 313 nm (ε = 21.927 M^−1^cm^−1^).

*(E)-1-(4-methoxyphenyl)-2-methyl-3-phenylprop-2-en-1-one* (**7c**): ^1^H-NMR (300 MHz, CDCl_3_): *δ* (ppm): 7.81 (d, *J* = 8.8 Hz, 2H, H-3, H-3‘), 7.42–7.39 (m, 4H, H_ar_), 7.35–7.31 (m, 1H, H_ar_), 7.10 (d, *J* = 1.5 Hz, 1H, H-9), 6.95 (d, *J* = 8.8 Hz, 2H, H-3, H-3‘), 3.85 (s, 3H, H-1), 2.25 (d, *J* = 1.5 Hz, 3H, H-8), ^13^C-NMR (75 MHz, CDCl_3_): *δ* (ppm): 199.7 (s, C-5), 160.1 (s, C-12), 142.8 (d, C-8), 139.0 (s, C-4), 135.0 (s, C-6), 131.5 (d, C_ar_), 131.5 (d, C_ar_), 129.5 (d, C_ar_), 128.5 (s, C-9), 128.2 (d, C_ar_), 114.1 (d, C-11, C-11′), 55.4 (q, C-13), 14.5 (q, C-7); UV/vis (10^−5^ M in CH_3_CN): λ_abs_ (max) =222 nm (ε = 16.828 M^−1^cm^−1^), 285 nm (ε = 25.217 M^−1^cm^−1^).

*(E)-Methyl-4-(2-methyl-3-oxo-3-phenylprop-1-en-1-yl)benzoate* (**7d**): According to GP, 4.03 g (30.0 mmol, 1.0 eq.) propiophenone (**5a**) and 4.92 g (30.0 mmol, 1.0 eq.) methyl 4-formyl benzoate (**6c**) were converted to methanol. The product **7d** was obtained as yellow oil after two column chromatographic purifications (1st column: 10:1, cHex:EtOAc; 2nd column: 10:1, cHex:EtOAc). Yield: 1.51 g (5.39 mmol, 18%), R_f_: 0.36 (10:1, cHex:Et_2_O), ^1^H-NMR (500 MHz, CDCl_3_): 8.07 (d, *J* = 8.4 Hz, 2H, H-11, H-11′), 7.77 (dd, *J* 8.3 Hz, 1.45 Hz, 2H, H-3, H-3′), 7.55 (m, 1H, H-1), 7.47 (dd, *J* = 8.0 Hz, 3.44, 4H, H-2, H-2′, H-10, H-10′), 7.17 (d, J = 2.1 Hz, 1H, H-8), 3.93 (s, 3H, H-14), 2.26 (d, *J* = 1.6 Hz, 3H, H-7).^13^C-NMR (75 MHz, CDCl_3_): *δ* (ppm): 199.0 (s, C-5), 166.6 (s, C-13), 140.3 (s, C_ar_), 140.2 (d, C-8), 138.7 (s, C-9), 138.0 (s, C-4), 132.0 (d, C-1), 129.8 (s, C_ar_), 129.7 (d, C_ar_), 129.6 (d, C_ar_), 129.5 (d, C-3, C-3‘), 128.3 (d, C_ar_), 52.3 (q, C-14), 14.7(q, C-7), FT-IR (ATR): ṽ (cm^−1^) = 1720.6 (s), 1647.4 (m), 1607.3 (w), 1435.0 (m), 1410.3 (w), 1357.3 (w), 1278.4 (s), 1261.9 (s), 1181.4 (m), 1109.3 (m), 1012.2 (m), 966.3 (w), 911.7 (w), 858.8 (w), 811.3 (w), 769.2 (m), 739.6 (s), 713.0 (s), 700.0 (s), 623.0 (w), 642.7 (m), 629.0 (m), 613.4 (m), 605.1 (w), 600.4 (w), 522.5 (w), HR-MS (ESI): m/z [M + H]^+^ 281.11744 (+ 078 ppm), [M + Na]^+^ 303.09933 (+ 0.53 ppm); UV/Vis (10^−5^ M in CH_3_CN): λ_abs_ (max) = 289 nm (ε = 24.117 M^−1^cm^−1^).

*(E)-Methyl-4-(3-(4-methoxyphenyl)-2-methyl-3-oxoprop-1-en-1-yl)benzoate* (**7e**): According to GP 1 4.93 g (30.0 mmol, 1.0 eq.) 4-methoxypropiophenone (**5b**) and 4.92 g (30.0 mmol, 1.0 eq.) methyl 4-formylbenzoate (**6c**) were converted to methanol. After column chromatographic purification (10:1, cHex:EtOAc) the product **7e** was recovered as a colorless solid. Yield: 0.79 g (2.54 mmol, 8%), R_f_: 0.33 (4:1, cHex:EtOAc), ^1^H-NMR (500 MHz, CDCl_3_): 8.07 (d, *J* = 8.4 Hz, 2H, H-4, H-4′), 7.82 (d, *J* = 8.8 Hz, 2H, H-12, H-12′), 7.47 (d, *J* = 8.2 Hz, 2H, H-11, H-11′), 7.08 (s, 1H, C-9), 6.96 (d, *J* = 8.8 Hz, 2H, H-3, H-3′), 3.93 (s, 3H, H-1), 3.88 (s, 3H, H-15), 2.25 (d, *J* = 1.5 Hz, 3H, H-8).^13^C-NMR 198.0 (s, C-6), 166.8 (s, C -14), 163.1 (s, C-2), 140.6 (s, C-10), 139.0 (s, C-7), 138.2 (d, C-9), 132.2 (d, C-12, C-12′), 130.3 (s, C-5), 129.8 (d, C-4, C-4′), 129.7 (s, C-13), 129.5 (d, C-11, C-11′), 113.7 (d, C-3, C-3′), 55.6 (q, C-15), 52.4 (q, C-1), 15.3 (q, C-8). FT-IR (ATR): ṽ (cm^−1^) = 1722.0 (s), 1634.4 (m), 1595.4 (s), 1507.3 (m), 1458.0 (w), 1437.5 (m), 1416.1 (m), 1357.2 (w), 1312.3 (m), 1284.6 (s), 1191.5 (m), 1182.3 (s), 1168.7 (s), 1143.2 (s), 1111.5 (s), 1014.4 (s), 1003.4 (m), 967.8 (w), 933.7 (m), 918.0 (m), 840.4 (m), 802.5 (w), 771.0 (s), 754.2 (w), 693.1 (m), 623.4 (s), 612.9 (s), 601.8 (s), 529.3 (m), 511.7 (w), HR-MS (ESI): m/z [M + H]^+^ 311.12802 (+ 077 ppm), [M + Na]^+^ 333.10973 (+ 0.00 ppm); UV/Vis (10^−5^ M in CH_3_CN): λ_abs._ (max) = 291 nm (ε = 31.299 M^−1^cm^−1^).

Photochemical procedures: (a) UV-irradiations of chalcones for analysis of the E/Z-photostationary states were performed in a LuzChem CCP-4V photochemical reactor (Luzchem Research Inc., Ottawa, Canada) using 16 UV-A lamps (phosphor-coated mercury low-pressure lamps, 12 inch, 8 watt each) and 10^−4^ M solutions in CH_3_CN in 1 cm quartz cuvettes; (b) singlet oxygen conditions were investigated using 200 mM stock solutions of chalcones and 5 × 10^−4^ mol/L sensitizer (TPP) in CDCl_3_. As reaction vessels, 2 mL pyrex NMR tubes were irradiated with three white LEDs (Osram L-865 Lumilix, 30 W each, 6500 K, color code 865) under constant oxygen bubbling. As results of the direct (350 nm) and indirect (^1^O_2_-induced) E/Z-isomerizations, the following set of photostationary and induced isomerization states resulted (Table 1) [45]:

## 5. Computational Part

All computations are performed with GAUSSIAN 16 Revision B.01 [46]. The structures are localized using the TPSS functional [47] with the def2-TZVP basis set [48]. Grimme’s dispersion (D3) with Becke–Johnson damping (BJ) [49] is added. All functions are implemented in the GAUSSIAN 16 program package. The presented computed structures are generated with the program CYL view [50].

## Data Availability

The data presented in this study are available from this article.

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
