# Peer review of "New Photochromic α-Methylchalcones Are Highly Photostable, Even under Singlet Oxygen Conditions: Breaking the α-Methyl Michael-System Reactivity by Reversible Peroxybiradical Formation"

_molecules, 2021, doi:10.3390/molecules26030642_

Round 1

Reviewer 1 Report

The paper by Renner et al. reports an exceptionally interesitng case of singlet oxygen reactivity.

The experiments are carefully designed and conducted, their analysis is supported by the dft calculation and the conclusions are very logical.

There are, however some minor issues that need to be addressed, concerning the description of experimetal procedures and data presentation.

  1. Scheme 5 shows the spectral changes recorded for illumination of one of chalcones. No data on the light source (type of the source, light flux, stirring, presence of oxygen) and conditions (solvent, concentration, temperature) are given.
  2. No characteristics of the main light source ued in this study is given: light flux, spectral characteristics of the source (white LEDs may have diverse emission spectra, depending on manufacturer and spectral temperature), and illumination geometry (NMR tubes are not ideal photoreactors - optical chatacteristics of their material should be givel as well, due to possible filter effect).
  3. Scheme 7b shows the reaction in the presence of chalcone. The descrition indicates photoisomerization of the calcone molecule, however this is not shown in the figure.

Otherwise the paper is very well written and I recommend its publication upon minor revision.

Author Response

Response.

We thank the reviewer for his/her very positive comments. The issues mentioned by the reviewer are addressed as follows:

  1. in Scheme 5, we have now included (to the legend) information on the experimental setup which we describe then in more detail in the experimental part included all data concerning the photo-stationary E/Z-ratios as well as the 1O2-induced E/Z-ratios of all compounds investigated.
  2. in the experimental part, we have now included information on the experimental setup which includes reaction chambers, concentrations, reactors and the two different light sources for direct excitation and 1O2-sensitization (mercury LPL and Osram white 865 color code lamps).
  3. in the legend to Scheme 7, we have now included "only E-7a is shown as the major product from a E/Z product mixture” because it appears misleading if we draw only Z-7a as the product structure.

Reviewer 2 Report

Interesting work for people involved in photosynthesis.

The reviewer has no comments to the experimental part.

Nevertheless, he/she pays attention to the use of theoretical calculations. In the presented version, the "confirmation" of the research hypothesis is based on speculations rather than on theoretical calculations, which are possible for this type of systems and for description of this type of phenomena. The reviewer refers to the source work written by J.N. Harvey, M. Aschi, H. Schwarz, W. Koch, Theor. Chem. Accounts 99 (1998) 95–99 or direct applications, e.g. in the work of Journal of Inorganic Biochemistry 173 (2017) 28–43.

Please comment on the possibility of localization of transition state associated with electron transfer using the Minimum Energy Crossing Point (MECP) described in the literature. If such calculations are possible, I suggest doing them in order to support the experiment.

Author Response

Response.

The reviewer points to a crucial point of the quenching process. Clearly, any singlet intermediate formed in the primary interaction have to undergo intersystem crossing to a triplet species that can eliminate triplet oxygen.

The issue mentioned by the reviewer is addressed as follows:

The Minimum Energy Crossing Point (MECP) method which is addressed by the reviewer, could not be calculated directly with the Gaussian program. This method finds the point of lowest energy where singlet and triplet potential hypersurfaces approach each other / cross. In our case we have compared singlet and triplet energies (which were very close in energies with <0.5 kcal/Mol) but whether singlet/tripet transition occurs cannot be determined by the theoretical reaction profile that we have used. One might assume that intersystem crossing is very rapid in these flexible and nearly energetically degenerated 1,4-biradicals with severe SOC contribution. But clearly, we have not clue from the calculations how fast ISC really is and high strong SOC and HFC contribution actually are.

We added the paragraph on page 8:

The calculated energy differences between the singlet and triplet biradical pathways in Scheme 9 are minimal (<0.5 kcal/Mol, not included in Scheme 9) and thus rapid intersystem crossing from the initial biradicals BR1/BR2 into the triplet manifold is expected to occur rapidly.

Reviewer 3 Report

In their article, the Authors describe the singlet oxygen induced isomerization of alpha methylated chalcone derivatives. Overall, I think it is a very well-conducted study, covering a nice combination of synthetic chemistry, spectroscopy and theoretical calculations. In my opinion, the Article may attract interest from the researcher of this field, and the manuscript can be accepted after the correction of the following minor remarks:

The exact E/Z ratios for the reactions of 7a-e under direct photochemical (page 3, Scheme 4) and singlet oxygen reaction conditions are missing (page 5, Scheme 6).

The General procedure of the direct photochemistry and photooxygenation of chalcones 7 are missing from the Experimental part.

Yields of 7a-c are missing from the Experimental

Although, the UV and NMR spectra of the reaction mixtures suggest the formation of the corresponding Z isomers, it would be nice to prepare at least one Z-7 product from a corresponding reaction mixture, in order to verify its structure by NMR and MS.

‘Conclusions’ part is missing.

Author Response

Response.

We thank the reviewer for his/her positive comments. The issues mentioned by the reviewer are addressed as follows:

  1. a) at the end of the experimental part, we have now included the exact E/Z ratios for the reactions of 7a-7e under direct photochemical and singlet oxygen reaction conditions. Because this set of data is not necessary for the mechanistic discussion, we have not included all data in the major text part but collected in a tabular collection in the experimental part (page 11);
  2. b) at the end of the experimental part, we have now included additional information on the experimental setup which includes reaction chambers, concentrations, reactors and the two different light sources for excitation (page 11);
  3. c) the chemical yields of 7a-7c were included into the from the experimental part (page 9);
  4. d) the reviewer is right and we will present these results with an extensive discussion of the photochromic behaviour, E/Z-structures, photophysics and quantum yields of forward and backward photoswitching in an additional full paper that we are writing currently (and citing now as ref. 46: Öngel, B.; Brüllingen, E.; Griesbeck, A. G. unpublished results).
  5. e) an additional conclusion part is now included as section Conclusion (page 9).

Round 2

Reviewer 2 Report

I accept the explanations presented by the authors of the work. It is obvious that the The Minimum Energy Crossing Point (MECP) calculation cannot be done in the Gausian program. The question from previous review concerned the potential possibilities of making such calculations  and the assessment whether the cost of these calculations is worth the results that can be obtained.

I propose to add to the manuscript the information that such calculations are possible and what information can be obtained from this type of calculations. 

Author Response

We agree with the reviewer points concerning the calculation of the energies and geometries of the crossing points between the two potential energy surfaces.

We have now included the sentence on page 8:

The calculated energy differences between the singlet and triplet biradical pathways in Scheme 9 are minimal (<0.5 kcal/Mol, not included in Scheme 9) and thus intersystem crossing from the initial biradicals BR1/BR2 into the triplet manifold is expected to occur rapidly. The computational analysis of these spin-forbidden processes can be performed by minimum energy crossing point (MECP) calculations.[44,45]

and the two new references:

  1. 4 Harvey, J. N. Spin-forbidden Reactions: Computational Insight into Mechanism and Kinetics, WIREs Comput. Mol. Sci. 2014, 4, 1-14
  2. Harvey, J. N.; Aschi, M.; Schwarz, H., Koch, W. The Singlet and Triplet States of Phenyl Cation. A Hybrid Approach for Locating Minimum Energy Crossing Points between Non-interacting Potential Energy Surfaces. Theor. Chem. Acc. 1998, 99, 95-99.
